# Human butyrylcholinesterase in Cohn fraction IV-4 purified in a single chromatography step on Hupresin

**Lawrence M. Schopfer**[1]*, **Emilie David**[2], **Steven H. Hinrichs**[3], **Oksana Lockridge**[1]

**1** Eppley Institute, University of Nebraska Medical Center, Omaha, Nebraska, United States of America, **2** CHEMFORASE, Mont-Saint-Aignan, France, **3** Department of Pathology and Microbiology, University of Nebraska Medical Center, Omaha, Nebraska, United States of America

* lmschopf@unmc.edu

**Data Availability Statement:** All relevant data are within the paper and its Supporting Information files.

## Abstract

Protection from the toxicity of nerve agents is achieved by pretreatment with human butyryl-cholinesterase (BChE). Current methods for purifying large quantities of BChE from frozen Cohn fraction IV-4 produce 99% pure enzyme, but the yield is low (21%). Our goal was to simplify the purification procedure and increase the yield. Butyrylcholinesterase was extracted from frozen Cohn fraction IV-4 in 10 volumes of water pH 6. The filtered extract was pumped onto a Hupresin affinity column. The previously utilized anion exchange chromatography step was omitted. Solvent and detergent reagents used to inactivate lipid enveloped virus, bacteria and protozoa did not bind to Hupresin. BChE was eluted with 0.1 M tetramethylammonium bromide in 20 mM sodium phosphate pH 8.0. BChE protein was concentrated on a Pellicon tangential flow filtration system and demonstrated to be highly purified by mass spectrometry. A high pump rate produced protein aggregates, but a low pump rate caused minimal turbidity. Possible contamination by prekallikrein and prekallikrein activator was examined by LC-MS/MS and by a chromogenic substrate assay for kallikrein activity. Prekallikrein and kallikrein were not detected by mass spectrometry in the 99% pure BChE. The chromogenic assay indicated kallikrein activity was less than 9 mU/mL. This new, 1-step chromatography protocol on Hupresin increased the yield of butyrylcholinesterase by 200%. The new method significantly reduces production costs by optimizing yield of 99% pure butyrylcholinesterase.

## Introduction

Human butyrylcholinesterase (BChE) in blood protects from the toxicity of nerve agents and organophosphorus pesticides by inactivating these poisons before they reach the nervous system. Animals pretreated with purified BChE are protected from lethal doses of soman, sarin, and VX [1–3] with no signs of toxicity, cognitive deficits, or behavioral decrements in comparison to animals not pretreated with human BChE.

BChE purified from human plasma is a tetramer with a half-life in the circulation of 2 weeks [4]. Purified BChE is stable for up to 20 years when a sterile, concentrated solution (e.g. 10 mg/mL) is stored at 4˚C [5, 6].

**Funding:** This work was supported by contract HDTRA1-14-1-0056 (to SH) from the United States Defense Threat Reduction Agency; URL = https://www.dtra.mil; and by the Fred & Pamela Buffet Cancer Center Support Grant P30CA036727 from NIH; URL = https://www.nih.gov;. CHEMFORASE thanks The French Ministry of Higher Education and Research, Bpifrance, Normandie Incubation and Réseau Entreprendre Normandie Seine for financial support; URL = https://www.normandie-incubation.com/en/offre/incubation-en/bpi-france-2. Emilie David, President of the CHEMFORASE Company, synthesized Hupresin. The funders had no role in study design, data collection and analysis, decision to publish, or preparation of the manuscript. The findings and conclusions in this article are those of the authors and do not necessarily represent the views of the U.S. Defense Threat Reduction Agency or the U.S. Department of Health and Human Services. Use of trade names is for identification only and does not imply endorsement by U.S. or French government agencies.

**Competing interests:** The authors have declared that no competing interests exist.

**Abbreviations:** BChE, butyrylcholinesterase, accession P06276; LC-MS/MS, liquid chromatography tandem mass spectrometry; PBS, phosphate buffered saline (12 mM sodium/potassium phosphate pH 7.4 plus 13 mM sodium chloride, 2.5 mM potassium chloride); PK, prekallikrein, accession P03952; PKA, prekallikrein activator, Hageman Factor, coagulation Factor XII, accession P00748; S/D, solvent detergent; SDS, sodium dodecyl sulfate; TMA, tetramethylammonium bromide; MWCO, molecular weight cut-off.

Procedures to manufacture large quantities of 99% pure human BChE from frozen Cohn fraction IV-4 have encountered problems of high cost, low yield, and insoluble protein aggregates in concentrated BChE preparations. Removal of prekallikrein has also been a concern. In a previous report [5] we described a purification method that started with extracting frozen Cohn fraction IV-4 with water at pH 4.5, pumping it thru a depth filter, and onto a Q-ceramic anion exchange column. Up to 70% of the starting BChE was lost in this step because much of the BChE passed through the Q-ceramic without binding. Q-ceramic was followed by affinity chromatography on Hupresin. BChE was eluted from Hupresin with 0.1 M tetramethylammonium bromide (TMA, Tokyo Chemical Industry, cat# T0135, Tokyo Japan) in 20 mM sodium phosphate pH 8. The final product was 99% pure, but the overall yield of 21% was unsatisfactory.

In the present report we describe a method that reduces production costs and preparation time, and increases the yield of 99% pure BChE. An important factor in the purification process is the nature of the starting Cohn fraction. We began exploring alternative approaches to purification when the commercial manufacturer changed their process for recovery of clotting factors in 2015 leading to Cohn fraction IV-4. The new commercial process generated a Cohn fraction IV-4 with different properties, possibly due to the use of a filter aid in the fractionation process. With the new process Cohn paste, it was possible to extract BChE into water at pH 6 and obtain a nearly clear supernatant after letting the solids settle out. In paste prior to 2015, the pH of the extracted supernatant had to be reduced to 4.5 before substantial settling and clarification occurred. The settled supernatant from the new process did not clog the depth filter during removal of fines as readily as the supernatant from the old process. This reduced the expenditure on filters. The new process filtered supernatant could be pumped directly onto the Hupresin column. This eliminated the need for the Q-Ceramic chromatography, which further reduced costs and time of the preparation.

Previously, Q-Ceramic anion-exchange chromatography was used to remove the solvent and detergent reagents that inactivate lipid enveloped viruses and bacteria. With our new process, we found that the solvent/detergent reagents were removed when the filtered extract was passed through Hupresin.

The most significant benefit of omitting anion exchange chromatography was the fact that the yield of 99% pure BChE increased by 200%, from 21% to 46%.

Additional improvements were necessary in the concentration and buffer exchange step for the purified BChE. Tangential flow filtration using a Pellicon 2 Mini cassette produced insoluble clumps of aggregated BChE protein when the BChE solution was pumped at a high rate. When the pump rate was slowed, this problem was minimized. Ultimately, the purified BChE was passed thru a 0.22 micron filter which removed residual particles and produced a clear solution.

An important consideration in the generation of purified BChE is the possible presence of prekallikrein and prekallikrein activator. These proteins may cause blood coagulation and hypotension when infused rapidly into the circulation. Analysis of Cohn fraction IV-4 by LC-MS/MS identified prekallikrein and prekallikrein activator. However, BChE purified by affinity chromatography on Hupresin contained no detectable prekallikrein and prekallikrein activator when analyzed by LC-MS/MS. A chromogenic assay for kallikrein activity was used to examine levels in six different lots produced over 2 years and showed only minimal kallikrein amidase activity.

## Materials

### Frozen Cohn fraction IV-4

Frozen Cohn fraction IV-4 was manufactured by Grifols Therapeutics Inc. (Clayton, NC). The material used in this study was obtained after 2015 when the company began using a new modification of its production process. The Certificate of Release reported that the plasma

pool was negative for hepatitis B antigen, HIV-1/2 antibody, HIV-1, hepatitis C virus, hepatitis B virus and donors were negative for syphilis. It was also negative for hepatitis A virus using FDA licensed methods. The viral load of parvovirus B19 did not exceed $10^4$ IU/mL. All plasma units in the pool were 100% inspected U.S. Source Plasma collected in FDA approved centers. Frozen paste was stored at -40˚C at Grifols for 1 month during preparation of the Certificate of Release. After 1 month the paste was trucked at -40˚C to the University of Nebraska Medical Center, where it was stored at -80˚C. The BChE in 80 kg of frozen Cohn paste originates from 8000 donor plasma units.

The paste was received as irregular small pellets (Fig 1) and dispersed in 10 volumes of cold water at pH 6. Solids settled to a hard layer at the bottom of a 1000 L mixing tank. After solids settled out of the paste suspension, the supernatant was relatively clear. The Zeta Plus filter was re-used to filter at least 4000 L of extract.

## Ultrapure water

All reagents were prepared with Ultrapure water. The Millipore ultrapure water system was described in detail in a previous publication [5]. Containers and tanks were washed with ultrapure water and were used exclusively for BChE procedures.

## The BChE purification facility

The purification process was conducted in a dedicated 5˚C cold room and an adjoining 22˚C preparation room. The cold room houses a 1000 L stainless steel mixing tank for extraction of the Cohn paste, a large mixer, a 1000 L plastic tank for storing cold ultrapure water, a Zeta Plus E16 dual zone depth filter (3M Purifications Inc., Meriden CT, cat# E16E07A90SP05A) and a 16EZA filter holder (3M Purifications Inc., Meriden CT), several Masterflex peristaltic pumps (Cole Parmer, Vernon Hill IL model 7528–10, cat# EW-07528–10) equipped with Easy Load II pump heads, (Cole Parmer, Vernon Hill IL cat# 77200–60) and Pharma Pure L/S18 tubing (Cole-Parmer, Vernon Hill IL, 5/16 inside diameter 7/16" outside diameter, cat# 742948000), 500 L and 50 L plastic tanks, 4.2 L of Hupresin (CHEMFORASE, Mont-Saint-Aignan, France; contact emilie.david@chemforase.com) packed into a 27 L acrylic QuikScale column (Millpore/Merck, Burlington MA, 250 mm ID x 550 mm, cat# GS251511) a Pellicon 2 Mini cassette filter (Millipore/Sigma, Burlington MA, 30 kDa pore, Screen type C, Biomax polyethersulfone membrane, 88cm$^2$ @ 0.11 M$^2$, cat# SK1M012W01) with a Pellicon Mini Cassette Holder (Millipore/Sigma, Burlington MA, cat# XX42PMIN). The holder compresses the cassette between a manifold plate that conveys fluids in and out of the cassette and an end plate. The polyethersulfone membrane has a cutoff of 30 kDa.

The room temperature preparation room houses the Millipore water purification system as well as balances, pH meter, -80˚C freezers, 4˚C refrigerators, biosafety cabinet for filter sterilization of concentrated BChE, sterile plastic supplies, salts, disposable gloves, and extra Zeta Plus filters. The preparation room has a floor drain, 2 sinks, and standard laboratory furniture.

Procedures performed in a location other than the BChE purification facility included BChE activity assay, measurement of protein concentration, gel electrophoresis, chromogenic endotoxin assay, LC-MS/MS, and the chromogenic kallikrein activity assay.

## Extraction of BChE from Cohn fraction IV-4

Seven hundred and twenty liters of ultrapure water containing 1 mM EDTA (Sigma-Aldrich cat# E2,629–0, St Louis, MO) were cooled to 5˚C in a 1000 L stainless steel tank. The rationale for adding EDTA was to inhibit bacterial growth. Eighty kilograms of new process Cohn fraction IV-4 at -80˚C were dispersed in cold water with the aid of a mixer. The Cohn paste was

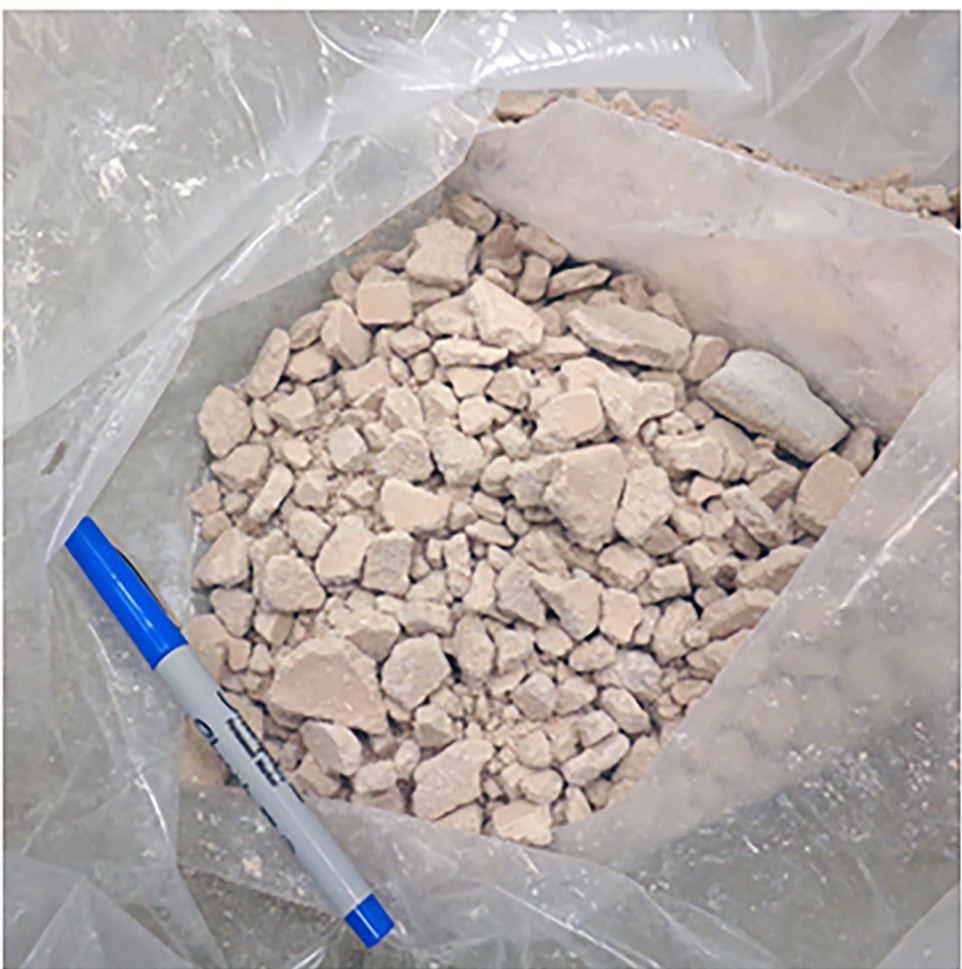

**Fig 1. New process frozen Cohn fraction IV-4 stored in plastic bags at -80˚C.** Gravel sized 1–3 cm pellets have a pink color.

taken directly out of a -80˚C freezer and added to the cold water. This avoided loss of BChE activity that occurred when paste was stored at -20˚C. The mixing time was 2 h at 60 rpm. The final suspension was at pH 6, measured with pH test strips (Sigma, St Louis MO, cat# P4536). Solids were allowed to settle out for 2 days at 5˚C. At pH 6, solids settled into a hard pack leaving BChE in the relatively clear supernatant. The conductivity of the clear supernatant was about 0.85 mS/cm. About 20% of the BChE was discarded in the solid hard pack.

## Filtration of Cohn extract

The clear brown extract was pumped through a Zeta Plus E16 dual zone depth filter installed in a model#16EZA filter holder. The extract was filtered to protect the Hupresin affinity column from the possibility that solids would accumulate on the Hupresin surface, blocking flow through the column.

The capacity of the Hupresin column (0.3 g BChE per L of Hupresin) limited loading to 250–300 L of the extracted supernatant volume. Consequently, some of the filtered supernatant was stored in 200 L holding tanks. The Masterflex pump rate for this process was set at 150 rpm yielding a flow rate of 36 L per h and normally a back pressure of 2 psi. The back

pressure rose as the filter plugged. The theoretical pressure limit for the Zeta Plus E16 filter is 50 psi. One hundred and fifty rpm is the practical limit for the Masterflex pump when using Pharma Pure L/S 18 tubing.

Some of the supernatant was pumped through the Zeta Plus filter and directly onto the Hupresin column. The maximum flow rate was determined by the pressure limit for the Hupresin column which was built on a Sepharose 4B matrix (Sigma, St Louis MO, cat# 4B200). The recommended pressure limit for Sepharose 4B is 1.14 psi. We set a pressure limit of 2 psi and limited the Masterflex pump rate to 60 rpm. Back pressure was monitored with a B-series over pressure cutoff switch (Ashcroft Inc., Stratford, CT, cat# B4-28-B-XFS) and an in-line sanitary pressure gauge (Ashcroft Inc., Stratford, CT, cat# 35-1036-SD15L-100).

## Solvent/Detergent treatment

A reference preparation was made by pumping 300 L of filtered extract directly onto the 4.2 L Hupresin column. A second 300 L of filtered extract was treated with solvent/detergent (0.3% tri-(n-butyl) phosphate, 1% Triton X-100) in two 150 L batches. Tri-(n-butyl) phosphate, 98% was from Alfa Aesar, cat# A16084, 98%, Wardhill, MA and Triton X100 was from Acros Organics/ThermoFisher cat# 9001 93–1, Pittsburg PA. Each batch was incubated for 2 days at 5°C before being pumped onto the 4.2 L Hupresin column. Chromatography details for the solvent/detergent treated extract are in Table 1.

## Hupresin

Hupresin was from CHEMFORASE, Mont-Saint-Aignan, France; contact emilie.david@chemforase.com. The ligand is a custom-synthesized hybrid of tacrine and huperzine, attached to Sepharose 4B through a 6-carbon linker. The structure of the ligand is illustrated in references [5, 7].

## Hupresin chromatography

An acrylic Quikscale column, 27 L capacity, was packed with 4.2 L of Hupresin. The Hupresin had been used 10 times previously for purifying BChE from frozen Cohn paste [5] and was stored in 2 M NaCl containing 0.01% sodium azide at 5°C. The Hupresin column was equilibrated with 28 L of freshly prepared 20 mM sodium phosphate pH 7.5, conductivity 2.4 mS/cm. At the end of equilibration, the conductivity of the buffer flowing out of the column was 2.4 mS/cm, the pH was 7.5, and the absorbance at 280 nm was less than 0.01. The buffer was pumped at a rate of 14 L per hour (Masterflex pump rate was set to 60 rpm), to avoid compacting the Sepharose 4B beads. The back pressure was monitored by an overpressure cut off switch that stopped the flow when the pressure exceeded 2 psi.

After the filtered Cohn extract was loaded, the Hupresin column was washed with 40 L of 20 mM sodium phosphate pH 8.0 and 30 L of 0.3 M NaCl in 20 mM sodium phosphate pH 8.0. At the end of the wash with 0.3 M NaCl, the pass through had an absorbance at 280 nm of less than 0.03. BChE was eluted with 12 L of 0.1 M tetramethylammonium bromide in 20 mM sodium phosphate pH 8.0. One-liter fractions were collected. BChE activity and absorbance at 280 nm were measured. Fractions with specific activity greater than about 490 units/mg served as the source of the 99% pure BChE. These fractions were pooled and concentrated with the Pellicon tangential flow filtration system.

The Hupresin column was washed with 10 L of 2 M NaCl in 20 mM sodium phosphate pH 8.0 to strip any residual protein. If the column was to be stored, 0.01% sodium azide was added to the stripping buffer. If another batch of supernatant was to be processed, the Hupresin column was equilibrated with 20 mM sodium phosphate pH 8.0 after stripping.

**Table 1. Solvent/detergent treated Cohn extract on 4.2 L Hupresin.**

| Sample [A] | Volume, L | A280 nm | Units | BChE mg[B] | u/ml | u/mg | |
|---|---|---|---|---|---|---|---|
| Before S/D | 300 | 13.5 | 887,800 | 1775 | 2.96 | | |
| Feed in S/D | 300 | 32.0 | 594,000 | 1190 | 1.98 | | |
| Pass through | 300 | 32.0 | 82,500 | 165 | | | Discard |
| Buffer wash | 10 | 14.8 | 9100 | 18 | | | Discard |
| Buffer wash | 30 | 0.293 | 0.024 | 1 | | | Discard |
| Buffer wash | end | 0.034 | 0 | 0 | | | |
| 0.3 M NaCl | 30 | 0.339 | 32,205 | 64 | | | Discard |
| 0.1 M TMA | 1 | 0.017 | 730 | 1 | | | Discard |
| 0.1 M TMA | 1 | 0.014 | 710 | 1 | | | Discard |
| 0.1 M TMA | 1 | 0.012 | 700 | 1 | | | Discard |
| 0.1 M TMA | 1 | 0.011 | 680 | 1 | | | Discard |
| 0.1 M TMA | 1 | 0.016 | 2430 | 5 | | | Discard |
| 0.1 M TMA | 1 | 0.490 | 130,000 | 260 | 130 | 477 | Save |
| 0.1 M TMA | 1 | 1.36 | 388,000 | 776 | 388 | 514 | Save |
| 0.1 M TMA | 1 | 0.334 | 93,700 | 187 | 93.7 | 505 | Save |
| 0.1 M TMA | 1 | 0.067 | 15,700 | 31 | 15.7 | 422 | Save |
| 0.1 M TMA | 1 | 0.028 | 5390 | 11 | | | Discard |
| 0.1 M TMA | 1 | 0.020 | 2860 | 6 | | | Discard |
| 0.1 M TMA | 1 | 0.017 | 2050 | 4 | | | Discard |
| 2 M NaCl | 10 | 0.030 | 31,500 | 63 | | | Discard |
| Total | | | | 1595 mg | | | |

A) Buffer wash is 20 mM sodium phosphate pH 8.0. Readings are given for 10 and 30 L pools. The entry designated Buffer wash end refers to readings made on the effluent after the 40 L had passed.

0.3 M NaCl indicates wash with 0.3 M sodium chloride in 20 mM sodium phosphate pH 8.0.

0.1 M TMA indicates elution with 0.1 M TMA in 20 mM sodium phosphate pH 8.0.

2 M NaCl indicates wash with 2 M sodium chloride in 20 mM sodium phosphate pH 8.0.

B) "BChE mg" was calculated from total units, using a specific activity of 500 units/mg.

## Pellicon concentration

A Pellicon 2 Mini Cassette tangential flow filter with a Biomax 30 membrane (30 kDa MWCO) was used for concentrating the peak fractions from the Hupresin chromatography. The purified BChE was pumped into the Pellicon at 5°C using a Masterflex pump fitted with PharmaPure tubing. The hold-up volume in the tubing and Pellicon was about 100 ml; this required the system to be flushed with about 100 ml buffer to recover concentrated BChE. We did not clear the lines by pumping air into the Pellicon because air caused BChE to form insoluble particulates.

BChE was concentrated with the pump rate set to the lowest possible speed of 6 rpm, producing 333 mL permeate per hour. High pump rates were avoided because they resulted in BChE protein aggregates that could not be completely removed by centrifugation or filtration. For every 1 L of permeate (discarded), the retentate volume was 2.2 L. The permeate had no BChE activity and no protein. Four liters of purified BChE eluate from the Hupresin chromatography were concentrated to 100 mL. The Pellicon and its plumbing were flushed with 200 mL of PBS. The resulting 300 mL were dialyzed to exchange the buffer, see the "Buffer exchange" section below. The concentrate was hazy.

## Buffer exchange

Two methods of buffer exchange were used.

The first method was used for 9 BChE preparations. Buffer was changed from 0.1 M TMA in 20 mM sodium phosphate pH 8.0 to PBS by diafiltration with the Pellicon 2. The protocol consisted of the following steps. 1) The volume was reduced to 50 mL. 2) PBS was added to dilute the TMA 5-fold. 3) The sample was concentrated to 50 mL and diluted 5-fold a 2nd time with PBS. This cycle of concentration, dilution, and concentration was repeated 7 times, reducing the final TMA concentration to 1 μM. The pump rate for the procedure was set to 80 rpm. This diafiltration regime produced a turbid solution of BChE in 100 mL (Fig 2A).

The second method was designed to minimize formation of aggregated BChE protein particles. BChE from 3–4 L of peak fractions taken from the Hupresin chromatography was concentrated in the Pellicon at a pump rate of 6 rpm to 300 mL, after flushing. The 300 mL of concentrated BChE was hazy (Fig 2B), but not turbid. The buffer was changed from 0.1 M TMA in 20 mM sodium phosphate pH 8.0 to PBS by dialysis in cellulose dialysis tubing (avg. flat width 43 mm, i.e. 1.7 in) (Sigma-Aldrich, St Louis MO, cat# D9527) against 2 x 20 L of PBS at 5˚C. PBS (phosphate buffered saline) was prepared in-house. A 10X solution contained 71.2 g $Na_2HPO_4{}^*2H_2O$, 9.6 g $KH_2PO_4$, 320 g NaCl, and 8 g KCl in 4 Liters. After 10-fold dilution this gave 140 mM Na/KCl in 12 mM $Na/KPO_4$ pH 7.4.

## Amicon stirred cell concentration

The dialyzed BChE was further concentrated in a 200 mL Amicon stirred cell ultrafiltration unit (Millipore/Sigma, Burlington MA, cat# UFSSC20001) using a PM10 membrane (Millipore/Sigma, Burlington MA, 10,000 MWCO, cat# PLGC16210) under a blanket of nitrogen.

## Filter sterilization and clarification

Concentrated BChE was filter sterilized inside a biosafety cabinet by pumping the BChE solution through a sterile 0.22 μm polyethersulfone membrane (Millipore/Sigma, Burlington MA, Sterivex GP sterile pressure driven filter unit with filling bell outlet, cat# SVGPB1010). The sterile, clear BChE solution was collected into a sterile container and stored at 4˚C.

A hazy BChE solution that had been concentrated in the Pellicon at a slow pump rate of 6 rpm could be filtered without plugging the 0.22 μm filter unit. In contrast, a turbid BChE solution had to be partially clarified by centrifugation and pre-filtration through a 0.45 μm membrane before it could be filtered through a 0.22 μm membrane.

## Endotoxin assay

Endotoxin levels were quantified with the kinetic chromogenic LAL assay (Charles River Laboratories, Wilmington MA, cat# R1708K). BChE was denatured in a boiling water bath for 3 min before measurement of endotoxin because native BChE hydrolyzes the chromogenic substrate to give artificially high endotoxin levels [8]. Heating inactivates BChE hydrolase activity without destroying endotoxin. The LSL assay substrate, Ac-Ile-Glu-Ala-Arg-p-nitroanilide, releases the yellow p-nitroaniline when hydrolyzed by BChE or by endotoxin-activated hydrolases in the Limulus Amebocyte Lysate component of the LAL assay kit.

## BChE activity assay

Activity was measured at 25˚C in 2 mL of 0.1 M potassium phosphate pH 7.0 in 1 cm path length quartz cuvettes containing 0.5 mM dithiobisnitrobenzoic acid (Sigma Cat# D8130, St Louis MO) and 1 mM butyrylthiocholine iodide (Fluka/Honeywell cat# 20820, Charlotte NC),

# A

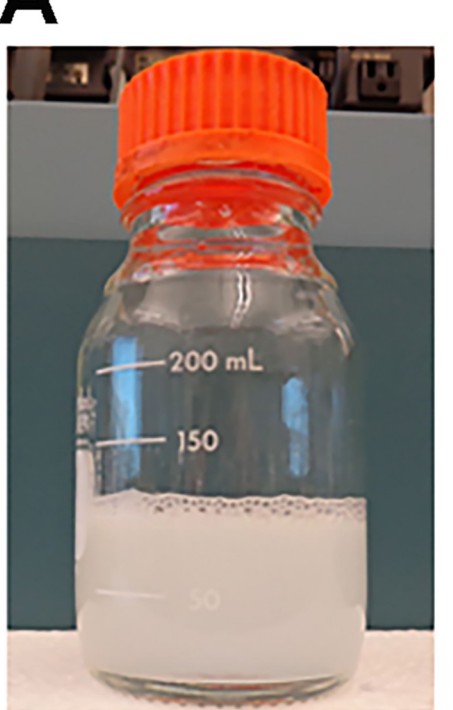

# B

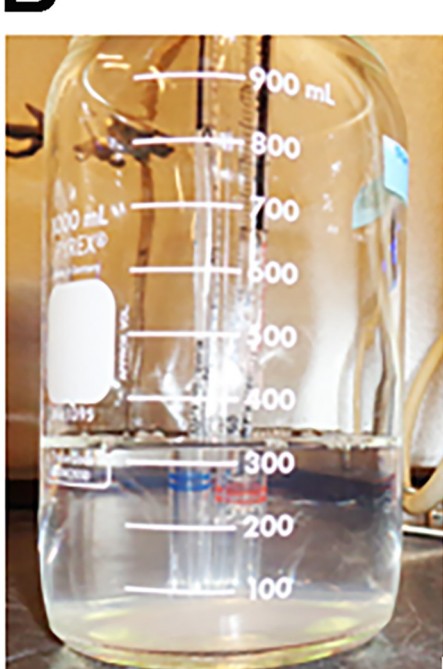

**Fig 2. Concentration and diafiltration in the Pellicon 2.** A) A pump speed of 80 rpm (9 L permeate per hour) was used for concentration and buffer exchange (diafiltration) of BChE in the Pellicon. 1.7 g of BChE in 5 L of 20 mM sodium phosphate pH 8.0 plus 0.1 M TMA was concentrated to 50 mL followed by diafiltration to yield 1.6 g of BChE in 0.1 L of PBS. The end-product was turbid. B) Using a pump speed of 6 rpm (333 mL permeate per hour) BChE was concentrated from 1.8 g in 4 L of 20 mM sodium phosphate pH 8.0 plus 0.1 M TMA. to 300 mL, after flushing. The end-product was hazy.

using a Gilford spectrophotometer interfaced to a MacLab computer. Increase in absorbance at 412 nm was recorded for 1 min and converted to μmoles per min using $E_{412nm} = 13,600\ M^{-1}\ cm^{-1}$. Units of activity are expressed as μmoles butyrylthiocholine hydrolyzed per min.

BChE that had been concentrated to 12,700 u/mL was assayed for activity after diluting the sample 2,000-fold as follows. Triplicate 20 μL aliquots were diluted into 1.98 mL of 1 mg/mL bovine albumin (Sigma, bovine fraction V, fatty acid free, cat# A8022, St Louis, MO) in phosphate buffered saline (PBS) to make triplicate 100-fold diluted samples. The 100-fold diluted BChE samples were further diluted by adding 50 μL to 950 μL of 1 mg/mL bovine albumin in PBS, for a final 2,000 fold dilution. The three 2,000-fold diluted BChE samples were assayed in triplicate using 10 μL of diluted BChE in a 2 mL reaction. Dilution into buffer containing albumin yielded 15% higher activity compared to dilution into buffer without albumin. Others have also found that addition of albumin to dilute solutions of BChE increases activity by minimizing sticking of BChE to the walls of the container [9, 10].

## BChE protein concentration

We determine the protein concentration of purified BChE in two ways.

First, we measured the absorbance at 280 nm in a Gilford spectrophotometer. A clear 1 mg/ml solution of purified human BChE in a 1 cm quartz cuvette has an absorbance at 280 nm of 1.8.

Second, we used the Pierce bicinchoninic acid protein assay (Thermo Scientific/Pierce, Waltham MA, cat# 23225). The bicinchoninic assay gives BChE concentration values that are 1.4-fold higher than the protein concentration calculated from absorbance at 280 nm. For example, a purified BChE solution in PBS with an absorbance of 1.8 at 280 nm corresponds to a protein concentration of 1 mg/ml. The same BChE solution measured with the bicinchoninic protein assay kit, using albumin as a standard, is calculated to have a protein concentration of 1.4 mg/ml.

## SDS polyacrylamide gel

Polyacrylamide 4–22% gradient slab gels, 0.75 mm thick, with a 4% stacking gel were poured in a Hoefer SE600 vertical slab gel apparatus (Hoefer Inc., Holliston MA). Samples were denatured in a boiling water bath for 3 min in the presence of dithiothreitol and SDS. Running buffer was 25 mM Tris, 180 mM glycine, and 0.01% sodium dodecyl sulfate. Molecular weight markers were Odyssey one-color protein molecular weight markers from LiCor (Lincoln NE) cat# 928–40000. Gels were stained with Coomassie blue.

## Nondenaturing polyacrylamide gel stained for BChE activity

Precast 4–20% polyacrylamide gels (BioRad, Hercules CA, cat# 4568094) were run at a constant voltage of 200 volts for 3.5 h in 22 mM Tris, 23 mM glycine pH 9. Human EDTA plasma (5 µl per lane) served as a reference for BChE C4 tetramers and BChE isozymes C1, C2, C3. Electrophoresis was stopped when the blue band for the albumin-bromophenol blue complex was 1 cm from the bottom of the gel, and the red band for hemoglobin was 2.5 cm from the bottom of the gel. Albumin and hemoglobin bands were visible in the plasma control lanes of the unstained gel. Bands with BChE activity were visualized by using the histochemical method of Karnovsky and Roots, adapted to polyacrylamide gels [11]. The substrate butyrylthiocholine iodide revealed brown-red bands for BChE.

## Trypsin digestion

Butyrylcholinesterase (180 µL of 2 mg/mL) from 8 preparations was reduced with 10 mM dithiothreitol and carbamidomethylated with 50 mM iodoacetamide. Samples were desalted by dialysis against 2 x 4 L of 20 mM ammonium bicarbonate pH 8 in Slide-a-Lyzer cassettes (Thermo Scientific/Pierce, Waltham MA, MWCO 7000, cat# 66373). A device with a molecular cutoff membrane of 7000 was selected to be sure the 76 kDa prekallikrein [12], if present, would be retained.

The volume of the dialyzed samples was reduced to 100 µL in a vacuum centrifuge to make the protein concentration 4 µg/µL assuming no losses during handling. Each 400 µg BChE sample was digested with 20 µg of trypsin (Sigma, St Louis MO, cat # T6667) at 37°C for 14 h. Trypsin was inactivated by addition of 1 µL of formic acid, reducing the pH to 3. Samples were centrifuged at 15,000 x g for 20 min before the top 85 µL were transferred to autosampler vials for mass spectral analysis.

## LC-MS/MS on the Orbitrap Fusion Lumos mass spectrometer

Trypsin-digested samples were analyzed on the Orbitrap Fusion Lumos mass spectrometer (Thermo Scientific, Waltham MA), a high-resolution instrument. A Thermo RSLC Ultimate 3000 ultra-high pressure liquid chromatography system (Thermo Scientific) was coupled to the Orbitrap Fusion Lumos mass spectrometer via an Acclaim PepMap 100 C18 trap column (Thermo Scientific, 75 µm x 2 cm, cat# 165535) and a Thermo Easy-Spray PepMap RSLC C18

separation column (Thermo Scientific, 75 μm x 50 cm with 2 μm particles, cat# ES803). A 2 μL sample (about 8 μg peptides) was loaded onto the trap column, washed with 100% solvent A (0.1% formic acid in water) for 10 minutes at 2 μL/min, shuttled onto the separation column, and eluted with a biphasic, linear gradient of 5 to 50% solvent B (0.1% formic acid in 80% acetonitrile) over 30 minutes followed by 50 to 100% solvent B over 40 minutes, at a flow rate of 0.3 μL/min. Parent ion mass spectra were collected in the Orbitrap detector (resolution of 120,000), in positive ion mode, with a charge state of 2–6, over a mass range of 350 to 1800 m/z. Mass tolerance was 10 ppm, data collection for a given mass was excluded after the first acquisition for 30 seconds. Maximum injection time was 100 msec, ion transfer tube temperature was 275˚C, ion spray voltage was 1900 volts, and automatic gain control was 400,000. Fragment ion spectra were taken using data dependent acquisition, isolation was in the quadrupole, and the detector was the Orbitrap (resolution 30,000). Fragmentation was by high energy collision-induced fragmentation at 35% normalized collision energy, maximum injection time was 60 msec, automatic gain control was 50,000, the scan range was auto normal.

### Search for prekallikrein (accession P03952) and prekallikrein activator (accession) P00748 proteins in mass spectrometry files

Mass spectrometry *.raw data files were converted to *.mgf files using MSConvert v 3.0 from Proteo Wizard. Data were loaded into Batch-Tag Web in Protein Prospector (http://prospector.ucsf.edu/prospector/mshome.htm) to search for peptide sequences in the purified BChE preparations that correspond to prekallikrein and prekallikrein activator. The search parameters included 1) Database SwissProt.2017.11.01. 2) Digest Trypsin. 3) Max 2 Missed cleavages. 4) Constant mods Carbamidomethyl. (C) 5) Variable Mods Oxidation (M). 6) Precursor Charge Range 2 3 4 5. 7) Masses monoisotopic. 8) Parent Tol 20 ppm, Frag Tol 30 ppm. 9) Instrument ESI-Q-high-res.

### Plasma kallikrein chromogenic activity assay

The plasma kallikrein activity colorimetric assay kit (BioVision, Inc, Milpitas CA, cat # K997 was used to measure kallikrein amidase activity in 6 purified human BChE preparations. The kallikrein activity in normal human pooled plasma is 220±30 mU/mL in the colorimetric assay performed with the BioVision Inc. kit.

## Results and discussion

### Solvent/Detergent passes through Hupresin without binding

Solvent/detergent (S/D) treatment is routinely used to inactivate lipid enveloped virus, bacteria and protozoa [13]. Lipid enveloped viruses include West Nile virus, influenza virus, HIV, and coronavirus. S/D reagents can be removed by anion exchange chromatography [14] under conditions that allow the protein of interest to bind, while the S/D reagents pass through without binding. We tested the possibility that BChE would remain bound to Hupresin in the presence of S/D and that S/D reagents would pass through Hupresin without binding.

We compared two 300-L samples from the same preparation. One sample was treated with S/D and the other was not.

First, we examined the disposition of Triton X-100 in the S/D treated sample. Triton X-100 has high absorbance at 280 nm ($E_{1\%} = 21$). The Cohn extract is brown and has an absorbance of 13.5 at 280 nm before addition of S/D. The absorbance was 32 after treatment with 1% Triton X-100/0.3% tri-(n-butyl)phosphate. The liquid that did not bind to Hupresin had an absorbance of 32 at 280 nm, indicating that Triton X-100 and the brown colored material in the

extract passed through during sample loading, without binding to Hupresin. In support of this conclusion, the effluent was visibly brown.

Solvent tri-(n-butyl) phosphate is an oily liquid that coelutes with Triton X-100 during anion exchange chromatography [14]. We therefore assume that tri-(n-butyl) phosphate coeluted with Triton X-100 during sample application on Hupresin. If any tri-(n-butyl) phosphate remained (MW 266.31 Da) it would be removed from BChE during the buffer exchange step in the Pellicon or during dialysis.

For purposes of clarity we will next describe the Hupresin chromatography of the S/D treated sample in detail, see Table 1. It can be seen that BChE activity drops from 3 u/ml to 2 u/ml upon addition of S/D reagents to the Cohn extract. This observation is consistent with reports that Triton X-100 inhibits the activity of human BChE (Ki = 13 μM) [15, 16]. Much of the lost activity was recovered in samples eluted with 0.1 M tetramethylammonium bromide (TMA) in 20 mM sodium phosphate pH 8.0, indicating that S/D does not prevent binding of BChE to Hupresin. Sixty-eight percent of the starting BChE eluted with 0.1 M TMA.

TMA is also a BChE inhibitor (Ki = 4 mM) [17]. However, activity assays of the TMA containing fractions were made under conditions where inhibition by TMA was minimal. That is, the activity of fractions containing 0.1 M TMA was measured by diluting 2 μL of the fraction into 2.0 mL of assay buffer, thereby diluting the TMA concentration 1000-fold to 0.1 mM. At 0.1 mM, the TMA concentration is 40-fold below Ki, therefore inhibition would be minimal.

Finally, we present a direct comparison of Hupresin chromatography for 300 L of extract not treated with S/D to 300 L of S/D treated extract, in Table 2. The 300 L aliquots were taken from the same preparation. S/D treated data are summarized from Table 1. Milligrams of BChE recovered in each step are effectively identical for both conditions. It can be concluded that there is no effect of 0.3% tri-(n-butyl) phosphate/1% Triton X-100 on the binding of BChE to Hupresin.

It may be difficult to completely remove all of the Triton X-10 from a protein. However, in those instances where Triton X 100 was used, there is no evidence for practical consequences from residual Triton X-100 on the final purified BChE. For example, Triton X 100 inhibits BChE [15] but the specific activity of the final BChE purified without or with solvent/detergent treatment was the same (see Table 2).

## Binding capacity of Hupresin for BChE in Cohn fraction IV-4 extract

Preliminary work suggested that approximately 0.3 mg of BChE could be loaded per mL of Hupresin. We tested this prediction for the preparations described in Tables 1 & 2., with and without S/D treatment. Milligrams of BChE per mL were calculated from units of activity per mL and a specific activity for BChE of 500 units per mg.

**Table 2. Comparison of BChE yield with and without solvent/detergent (S/D) in the Cohn extract.**

|  | No S/D | With S/D |
|---|---|---|
| Feed 300 L before S/D | 1775 mg BChE | 1775 mg BChE |
| Pass through, 300 L | 62 mg | 165 mg |
| Buffer wash, 40 L | 20 mg | 19 mg |
| 0.3 M NaCl, 30 L | 104 mg | 64 mg |
| 0.1 M TMA, 3 L | 1210 mg pure BChE | 1223 mg pure BChE |
| 0.1 M TMA 1 L side-fraction | 81 mg | 61 mg |
| 2 M NaCl 10 L | 119 mg | 63 mg |
| Total | 1596 mg | 1595 mg |

There were 2.96 u/mL BChE in the starting 300 L of Cohn extract before treatment with S/D, Table 1. This translates into a BChE protein concentration of 0.0059 mg/mL or 1776 mg of BChE in 300 L, Tables 1 & 2. We loaded the 300 L of S/D-treated extract onto the 4.2 L Hupresin column. A total of 370 mg of BChE were discarded, Table 1, of which only 165 mg failed to stick to the Hupresin. The 3 fractions with the highest specific activities contained a total of 1223 mg BChE. The side fraction with a specific activity of 422 u/mg contained 31 mg of BChE. Combining these two values indicates that 1254 mg of BChE bound to the 4.2 L of Hupresin, or 0.299 mg/mL.

An identical 300 L portion of filtered Cohn extract without added solvent/detergent was chromatographed on 4.2 L Hupresin. A total of 305 mg of BChE were discarded in the pass through, buffer wash, and sodium chloride washes (Table 2). The unbound pass-through fraction contained 62 mg which is 103 mg less than that from the S/D-treated sample, suggesting that the S/D reagents may slightly reduce BChE binding. In the end, both samples yielded 1.2 g of highly purified BChE, corresponding to a recovery 69% (1220 ÷ 1775) (Table 2).

It was concluded that the binding capacity for BChE was 0.3 mg/mL Hupresin and was the same with or without solvent/detergent in the Cohn extract. Amounts of BChE in excess of 0.3 mg/mL will bind to Hupresin, but the excess is washed off by 0.3 M sodium chloride. The 0.3 mg/mL limit on BChE binding to Hupresin dictates that 80 kg of Cohn fraction IV-4 be processed in 250–300 L portions on 4.2 L Hupresin in the one-step purification protocol. Scale up to bind the BChE in 80 kg of frozen new process Cohn paste would require 12 L of Hupresin. We could process the entire 80 kg of Cohn fraction IV-4 at once when using our old 2-step chromatography process because the amount of BChE recovered from the 30 L Q-Ceramic column was low enough that it did not saturate the Hupresin.

Highly purified BChE was obtained in a single chromatography step on Hupresin with a yield of 46% from extracts treated with solvent/detergent (S/D) as well as in extracts not treated with S/D.

## SDS gel electrophoresis

Samples from Table 1 were treated with dithiothreitol and SDS and loaded on an SDS gel. The Coomassie blue stained gel in Fig 3 shows an identical pattern of protein bands for solvent/ detergent treated Cohn extract in lane 1 and for the pass through Hupresin fraction in lane 2, indicating that most protein contaminants fail to bind to Hupresin. Contaminant proteins in Cohn fraction IV-4 were identified by LC-MS/MS. LC-MS/MS data were reported in Schopfer et.al [5]. The most abundant proteins are transferrin P02787 and albumin P02768. Other abundant proteins are hemopexin P02790, haptoglobin P00738, immunoglobulin, hemoglobin P69905, and ceruloplasmin P00450. A band for ceruloplasmin is not visible in Fig 3 lanes 1 and 2, but ceruloplasmin was highly visible as a blue, copper binding protein in our earlier anion exchange chromatography experiments [5]. Bands for BChE P06276 are not visible in lanes 1 and 2, but are present in fractions eluted with 0.1 M TMA, Fig 3 lanes 4, 5, and 6.

The 340 kDa BChE tetramer is a dimer of dimers with 4 identical subunits [18, 19]. The disulfide linked dimers are reduced by dithiothreitol to 85 kDa under denaturing conditions. A nonreducible dimer at 170 kDa is present in all purified BChE preparations.

## BChE purity

BChE purified from new process Cohn fraction IV-4 by a single chromatography step on Hupresin affinity resin is 99% pure. Purity was assessed by specific activity using butyrylthio-choline as substrate, where 500 u/mg protein was considered to be 99% pure. Purity was also

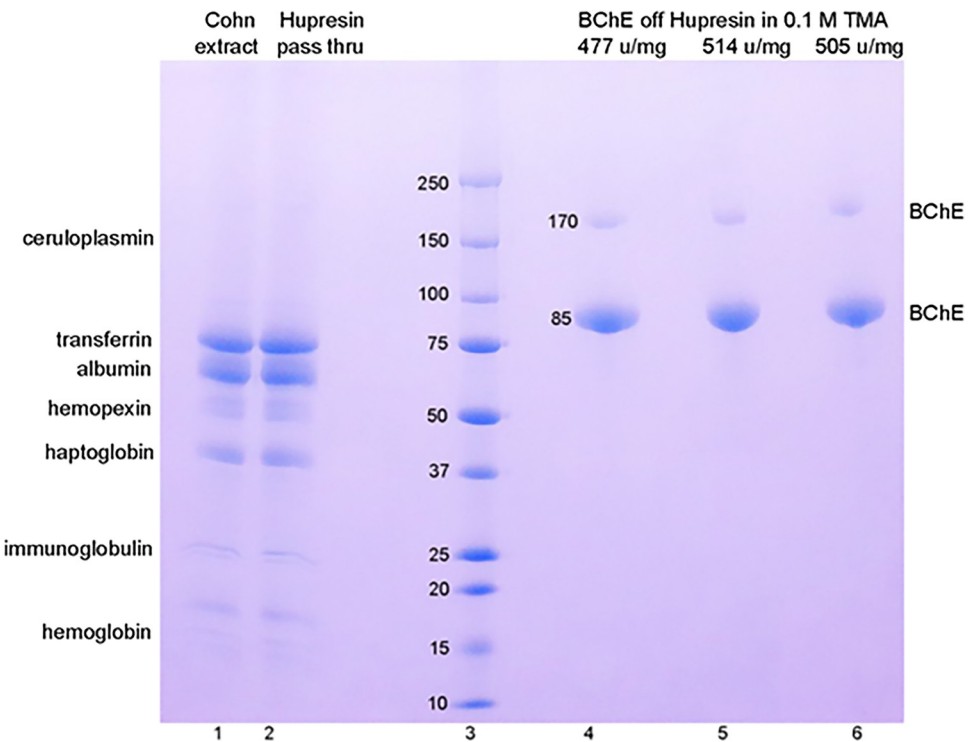

**Fig 3. SDS polyacrylamide gel stained with Coomassie blue.** Proteins were loaded at 5 μg per lane. Lane 1 Cohn extract. Lane 2 Hupresin pass through. Lane 3 molecular weight markers. Lanes 4, 5, and 6 highly purified BChE eluted from Hupresin with 0.1 M tetramethylammonium bromide in 20 mM sodium phosphate pH 8. In the presence of dithiothreitol and SDS, the 340 kDa BChE tetramer dissociates into 85 kDa monomers and a nonreducible dimer at 170 kDa. The most abundant proteins in the Cohn paste extract are listed.

assessed by contaminating bands seen in SDS PAGE stained with Coomassie blue (Fig 3), where no contaminating bands could be seen.

The polyproline peptide (PSPPLPPPPPPPPPPPPPPPPPPPLP) from RAPH1 (Ras-associated and pleckstrin homology domains-containing protein 1, lamellipodin, gi:82581557) that serves to lock-in the BChE tetramer [20] was identified in the mass spectral data.

## Tetramer status of BChE purified by chromatography on Hupresin

Nondenaturing polyacrylamide gel electrophoresis in the absence of SDS allows visualization of BChE tetramers, trimers, dimers, and monomers from plasma by staining for BChE activity. The human plasma sample in Fig 4 (lanes 1 and 10) has an intense band for the BChE tetramer (C4) and faint bands for BChE monomers (C1) and dimers (C2 and C3). The C2 band is a disulfide linked dimer between one subunit of BChE and one molecule of albumin [21]. The C3 band is a disulfide linked BChE dimer. Eight BChE samples purified from Cohn fraction IV-4 (lot# 61–78) have an intense band at the position of the BChE tetramer (C4) in Fig 4 lanes 2–9, but no bands for isozymes C1, C2, and C3. Lot #61 (Fig 4 lane 2) has a weak band at the position of a BChE trimer. It was concluded that BChE purified from Cohn fraction IV-4 by affinity chromatography on Hupresin consists predominantly of tetramers.

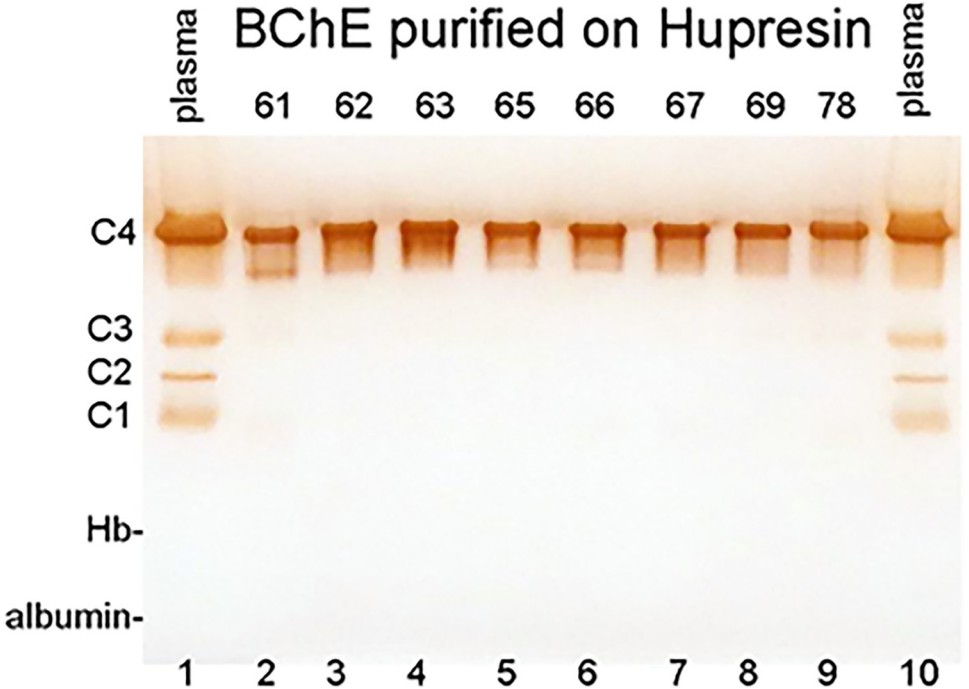

**Fig 4. Nondenaturing polyacrylamide gel stained for BChE activity.** The pattern of BChE active bands from human EDTA plasma (5 μL) in lanes 1 and 10 provides a well-established reference for the oligomeric state of BChE, where C1 is monomeric BChE at 85 kDa, C2 is a disulfide crosslinked dimer of BChE and albumin [21], C3 is the BChE dimer at 170 kDa, and C4 is tetrameric BChE at 340 kDa. The nondenaturing gel has an intense C4 band for the BChE tetramer and weak bands for BChE isozymes C1, C2, and C3. BChE purified Cohn fraction IV-4 with Hupresin lot# 61–78 in lanes 2–9 consists predominantly of tetramers. Purified BChE was loaded at 0.01 units (0.02 μg; 0.23 pmoles) per lane. A red band for hemoglobin (Hb) and a blue band for the albumin-bromophenol blue complex in plasma (lanes 1 and 10) were visible in the unstained gel. Electrophoresis was performed at 200 volts constant voltage for 3.5 h.

## BChE isozymes are unstable

BChE purified by affinity chromatography on Hupresin has almost no C1, C2, and C3 isozymes. These minor forms of BChE were present in Cohn extract, though at reduced levels compared to plasma. Previously, we have found that the minor forms are unstable. For example, if a nondenaturing gel is run at constant mAmp rather than at constant voltage, the gel gets hot and the minor forms are inactivated. This suggests the absence of C1, C2, and C3 isozymes in purified BChE preparations can be explained by inactivation of the isozymes during the purification procedure.

## Stability of BChE

BChE produced by various methods when stored in PBS, sterile and concentrated (e.g. 6000–7000 u/mL), at 4˚C, is stable for up to 20 years [22]. Stability was assessed by activity assays.

BChE lyophilized in 52 mM L-histidine pH 6.0 plus 240 mM sucrose, 0.3 mM Tween-20 and stored at 27˚C is relatively stable for 38 months (3 years) but ultimately loses 46% of its starting activity over 90 months (7.5 years). See Fig 5. We did not investigate the biochemical changes that occur in the lyophilized BChE protein during years of storage at 27˚C. Stability studies of stored antibodies suggest performance loss can be due to aggregation, fragmentation and oxidation [23].

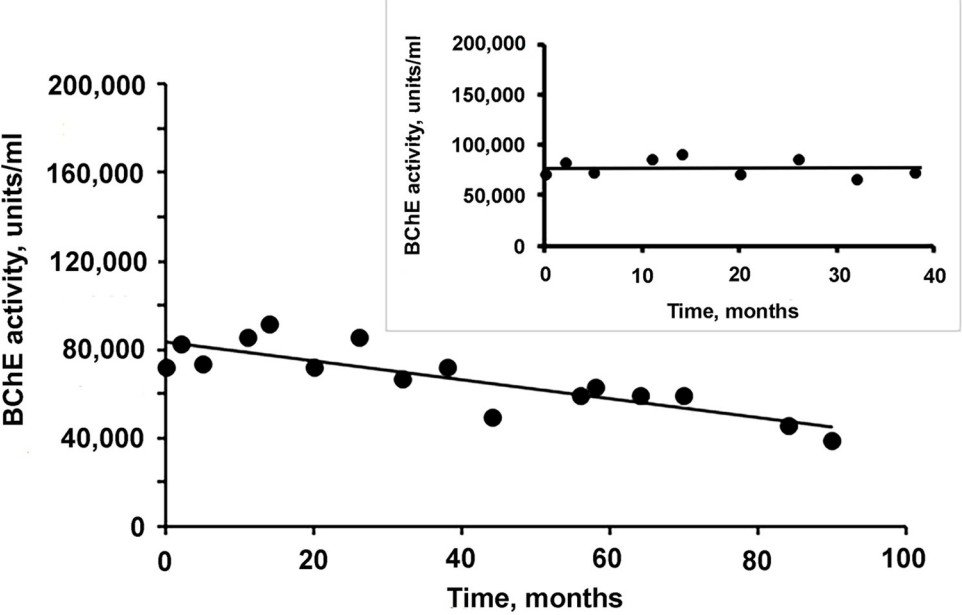

**Fig 5. Stability of lyophilized BChE.** BChE, purified from new Cohn fraction IV-4, was lyophilized in 52 mM L-histidine pH 6.0 plus 240 mM sucrose, 0.3 mM Tween-20, and stored at 27°C. 20 separate vials were made. Periodically, a vial was reconstituted in water and the activity of BChE measured.

When BChE in crude, new-process Cohn fraction IV-4 starting material is stored at -80°C, it is stable for at least 7 years. When new process Cohn fraction IV-4 was stored at -20°C for two years, it lost 33% of it starting activity. One 80 kg lot of Cohn fraction IV-4 was stored at -20°C for 3 days after the -80°C freezer broke. We processed the -20°C paste through our one-step purification protocol and got the disappointing result of 74% pure BChE. The faintly yellow 74% pure BChE solution contained contaminants at 45, 37, 25 and 12 kDa.

### Endotoxin

Proteins intended for administration to humans must have an endotoxin level less than 1 EU/ml to avoid toxic shock. We purify BChE in a non-GMP facility. The BChE we produce is suitable for animal and in vitro studies but is not suitable for use in humans. Despite the non-GMP status of our facility, the BChE we produce has a very low endotoxin level, less than 1 EU/ml. The use of Ultrapure water for all procedures helps to explain the low endotoxin level in our purified BChE preparations.

### Mass spectrometry analysis of BChE preparations purified on Hupresin found no prekallikrein (PK) or prekallikrein activator (PKA)

Possible contamination of purified BChE by prekallikrein (PK, accession #P03952) or prekallikrein activator (PKA, accession # P00748) is a concern because these proteins can be activated to induce blood coagulation [24] or to adversely reduce blood pressure [25]. Cohn fraction IV-4 contains PKA [26, 27] detected as kallikrein activity. We confirmed this result by LC-MS/MS analysis. However, there is no PK or PKA in Hupresin purified BChE. We searched for PK and PKA in our purified BChE preparations by LC-MS/MS on the Orbitrap Fusion Lumos mass spectrometer. No peptide attributable to either PK or PKA was found in any of the 8 preparations analyzed. The BChE preparations had been concentrated in the

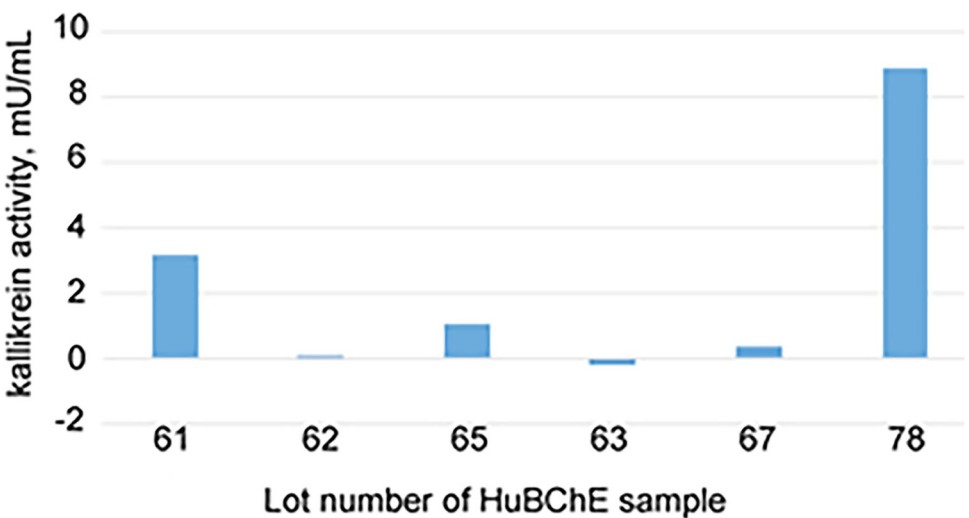

**Fig 6. Kallikrein activity in Hupresin-purified BChE preparations.** In a previous report we showed that human BChE hydrolyzes the chromogenic substrate used for measuring endotoxin concentration, Ac-Ile-Glu-Ala-Arg-p-nitroanilide [8]. The peptide substrate in the endotoxin assay is similar to that in the kallikrein activity assay (S-2302), but we found that BChE does not hydrolyze S-2302.

Pellicon with a 30 kDa cutoff membrane or in an Amicon stirred cell with a 10 kDa cutoff membrane. These membranes would have retained PK and PKA because the molecular weight of glycosylated prekallikrein is 80,000 [28] and that of prekallikrein activator is 76,000 [12].

## Chromogenic assay of kallikrein activity

We used a second method to analyze our purified BChE preparations for the presence of kallikrein. Kallikrein is produced from prekallikrein by the protease activity of prekallikrein activator (factor XIIa) [28]. Kallikrein is a serine protease. A standard assay for kallikrein activity measures release of the yellow product p-nitroaniline from the amide substrate Pro-Phe-Arg-p-nitroanilide (S-2302). Fig 6 shows kallikrein activity results for 6 lots of Hupresin-purified BChE. Activity values ranged from 0 to 9 µmoles per min per mL (mU/mL).

## Comparison of BChE yield from 2-step chromatography and 1-step chromatography

In our previous report we purified BChE from old process Cohn fraction IV-4 by anion exchange chromatography on 30 L of Q-ceramic followed by affinity chromatography on 4.2 L of Hupresin [5]. Up to 70% of the starting BChE passed through the anion exchange column without binding. An average of 1.8 g of highly purified BChE was recovered from the starting 8.6 g of BChE in 80 kg of Cohn fraction, for a yield of 21% (Table 3). The old process Cohn paste was not suitable for 1-step chromatography because solids settled out at pH 4.5 rather than at pH 8. BChE binds poorly to Hupresin at pH 4.5. When the pH 4.5 filtered extract was adjusted to pH 8, the yield and specific activity of BChE that eluted from Hupresin were unsatisfactory.

When the new process Cohn paste was extracted into water, the pH was 6. It yielded a clear supernatant after solids had settled out. Fines were removed by filtration without clogging the filter. BChE in the filtered extract bound to Hupresin in the absence of or in the presence of solvent/detergent. Solvent/detergent eluted from Hupresin in the buffer wash, with minimal

**Table 3. Yield of highly purified human BChE from 2-step chromatography and 1-step chromatography.**

| | Grams BChE in 80 kg paste [A] | Grams BChE after Q-ceramic | Grams BChE after Hupresin | % yield |
|---|---|---|---|---|
| 2-step chromatography old process Cohn paste | 8.6 | 2.6 | 1.8 | 21% |
| 1-step chromatography new process Cohn paste | 5.7 | --- | 2.64 | 46% |
| | 5.2 | --- | 2.4 | 46% |

loss of BChE (10 to 15%). The yield of BChE was 46% of the starting amount, representing a 200% improved yield compared to our previous protocol. The specific activity of highly purified BChE prepared by 1-step chromatography on Hupresin was 500 u/mg, similar to that from the 2-step chromatography procedure. 80 kg of frozen old process paste contained 8.6 g BChE, while 80 kg of new process paste contained 5.7 g BChE.

## Advantages and disadvantages of affinity chromatography on Hupresin

Purification of BChE from frozen Cohn fraction IV-4 by a single chromatography step on Hupresin has the following advantages. 1) There is no need for the expensive Q-ceramic anion exchange resin and associated Quickscale column. 2) The yield of purified BChE per 80 kg of frozen Cohn paste at 2.4 g out of 5.5 g of starting BChE is an improvement of 200% over the yield of 1.8 g out of 8.6 g starting BChE in the 2-step chromatography procedure. 3) The time to complete a run is shorter. 4) BChE purified on Hupresin has undetectable or minimal levels of PK and PKA. 5) Solvent/detergent reagents pass through Hupresin without binding, while BChE remains bound. 6) The specific activity of BChE purified in a single step on Hupresin is the same as for BChE purified in 2 chromatography steps. 7) The cost of purified human BChE is reduced by savings in supplies, time, and increased yield.

It is important to point out that purification of BChE from plasma in a single step with Hupresin yields BChE that is only 10–15% pure [29]. In order to obtain 99% pure BChE, a preliminary purification such as ion exchange chromatography on DEAE Sepharose or Q-Ceramic is necessary when the starting material is human plasma.

## What is needed to make Hupresin useful for large scale purification of BChE for human use

Purified human BChE intended for use in humans must be purified in a GMP facility with GMP quality reagents. In the year 2022, Hupresin is not available as a GMP-certified reagent. However, the method described here establishes the feasibility for industrial scale purification using Hupresin. The processing rate could be increased if the Hupresin ligand were attached to a noncompressible solid support that can withstand high pressure such as Profinity epoxide resin (back pressure limit 80 psi) (Biorad, Hercules CA, cat# 1560201) or Ceramic hyper DF (back pressure limit 1000 psi) (Pall Corp, Port Washington, New York). The new method significantly reduces production costs by optimizing yield and quality of 99% pure butyrylcholinesterase.

## Supporting information

**S1 Raw images.**
(PDF)

## Acknowledgments

Mass spectrometry data were obtained by the Mass Spectrometry and Proteomics Core Facility at the University of Nebraska Medical Center. Protein Prospector programs are available at no

cost. Protein Prospector Programs were developed in the University of California San Francisco Mass Spectrometry Facility, directed by Dr. Alma Burlingame,.

## Author Contributions

**Conceptualization:** Oksana Lockridge.

**Funding acquisition:** Steven H. Hinrichs.

**Investigation:** Lawrence M. Schopfer, Oksana Lockridge.

**Methodology:** Lawrence M. Schopfer, Oksana Lockridge.

**Project administration:** Steven H. Hinrichs.

**Resources:** Emilie David.

**Software:** Lawrence M. Schopfer.

**Supervision:** Oksana Lockridge.

**Writing – original draft:** Oksana Lockridge.

**Writing – review & editing:** Lawrence M. Schopfer, Emilie David, Steven H. Hinrichs, Oksana Lockridge.

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
