## [Decision Letter · Decision Letter 0]

8 Dec 2022

PONE-D-22-24553Human butyrylcholinesterase in Cohn fraction IV-4 purified in a single chromatography step on HupresinPLOS ONE

Dear Dr. Schopfer,

Thank you for submitting your manuscript to PLOS ONE. After careful consideration, we feel that it has merit but does not fully meet PLOS ONE’s publication criteria as it currently stands. Therefore, we invite you to submit a revised version of the manuscript that addresses the points raised during the review process.

ACADEMIC EDITOR: The minor revision is needed.

We look forward to receiving your revised manuscript.

Kind regards,

Ajaya Bhattarai

Academic Editor

PLOS ONE

https://journals.plos.org/plosone/s/fileid=ba62/PLOSOne_formatting_sample_title_authors_affiliations.pdf.

Additional Editor Comments:

One of the reviewers accepted the manuscript and another suggested for minor review. So, the Academic editor would like to see the revised work as soon as possible.

Reviewers' comments:

Reviewer's Responses to Questions

**Comments to the Author**

1. Is the manuscript technically sound, and do the data support the conclusions?

Reviewer #1: Yes

Reviewer #2: Yes

2. Has the statistical analysis been performed appropriately and rigorously? 

Reviewer #1: Yes

Reviewer #2: N/A

3. Have the authors made all data underlying the findings in their manuscript fully available?

Reviewer #1: Yes

Reviewer #2: Yes

4. Is the manuscript presented in an intelligible fashion and written in standard English?

Reviewer #1: Yes

Reviewer #2: Yes

5. Review Comments to the Author

Reviewer #1: This is a new method using 1-step chromatography protocol and innovative to reduce the production cost of pure butyrylcholinesterase (BChE) which may protect them from the toxicity of nerve agents and directly concern to the neuroimprovement.

Reviewer #2: Comment on PONE-D-22-24553

General:

The authors describe the purification of butyrylcholesterase (BChE) from Chon fraction IV in a single step chromatography using Hupresin as affinity matrix. BChE is an important human enzyme that can be used to treat poisoning with e.g. cocaine or other drugs. It is also required for the degradation of certain muscle relaxants, given in combination anesthesia. In addition, administration of BChE can protect to a certain degree to intoxication with organophosphates. Hence, there is a big pharmaceutical interest for BChE protein. Unfortunately, due to the complex structure of BChE (heavily glycosylated, forming a tetramer with the aid of an additional proline-rich peptide) it was so far not successful to produce BChE in a recombinant manner with comparative stability and activity properties as the protein purified from human plasma.

Hence, it is of great importance to develop a highly effective but nevertheless simple purification process from human plasma. Schoper et al. describe a simplified process for BChE purification in large scale (yet no GMP conditions) with higher yields. The BChE preparations have been checked against the most concerning contaminants: kallikrein and LPS / endotoxins. This is very well described.

The manuscript is described in a very propriate manner. The experimental procedure is described in great detail and the results are well presented and discussed.

Line 517: “contaminant proteins in Chon fraction IV were identified by LC-MS/MS

=> MS-Data for protein identification might be added as Supplementary Information.

Fig 4: Jugded only from the shown gel it can not be concluded that the protein is a tetramer, since there is no marker used. There are several high molecular weight marker available. Addition of a marker would be very helpful. The staining of the gel is derived from the activity of BChE. Tetramers , trimers, dimer and monomeric forms were active. However, in lane 2 (sample 62) there is a distinct band clearly visible below the main band, but this band does not correlate to the trimer, dimer or monomer form. This should at least be mentioned and discussed. Is it possible to stain the gel after the reaction with Coomassie blue?

Lane 572 PBS should be used for phosphate buffered saline. It is claimed that the BChE is stable in PBS at 4°C for 20 years. Please give more details on the origin of the samples or Refs.

Have the samples all been produced under the same conditions using same reagents ?

Stability assessment was done by activity measurements. Has the integrity of the proteins be checked ? e.g. by MS or analytical SEC? Samples shown in Fig 5 are all derived from the new Chon fraction correct ?

The BChE prepared from the new Chron fraction is less stable than the old batches (up to 20 years at 4°C). This should be mentioned and discussed ? Is there a possible reason for this ?

Kallikrein activity: There are certain differences between different BChe batches. What is the upper limit for kallikrein kinase activity allowed for human preparations – please add value – as done for LPS level (despite this material is not intended for use in humans).

Line 671 “… dos not yield pure protein” please give a number.

684. The last sentence is a little bit misleading.

Is the BChE produced with the new method 99% pure ? Please describe the method for estimating this rate.

More analytical data for the purified BChE should be added. E.g. analytical SEC, analytical HIC, MS-Data. HPLC…..Maybe it is possible to identify the proline-rich peptide.

As the protein has been washed with Triton x-100: How can you be sure, that Triton X-100 is completely removed and does not stuck to the protein ? I bet it does.

Minor topics:

Please give sources for “complex” compounds.

Line 176: Source for EDTA

Line 181: Please give conditions for mixing time (rpm, …)

Line 276 The abbreviation TMA is used. It should be introduced, when the term is used the first tim (line 76 tetramethylammonium bromide). Please give source.

Line 211: source for solvents and detergent (tri-(n-butyl) phosphate, Triton X-100)

Line 278: PBS Please explain: is this “self made” PBS (please give composition) or from an supplier (please give source)

Lines 318/319: Please give source for dithiobisnitrobenzoic acid, and butyrylthiocholine iodide.

Line 325 Source for “bovine albumin” which kind ? fatty acid free ?

346 SDS polyacrylamide: Which running buffer was used ? Which marker was used ? please give sources.

352: Nondenaturing polyacrylamide gel: Which running buffer was used ?

428: Citation should be in the reference section. Use number for ref. only

6. PLOS authors have the option to publish the peer review history of their article (what does this mean?). If published, this will include your full peer review and any attached files.

Reviewer #1: **Yes: **Dr. Shyam Narayan Labh

Reviewer #2: No

---

## [Author Response · Author response to Decision Letter 0]

19 Dec 2022

Reviewers comments and responses

Comment 1 Line 517: “contaminant proteins in Chon fraction IV were identified by LC-MS/MS

 => MS-Data for protein identification might be added as Supplementary Information.

Response 1 The following sentence was added to the first paragraph in the section entitled SDS Gel Electrophoresis. LC-MS/MS data were reported in Schopfer et.al [Schopfer, Lockridge, David & Hinrichs, Purification of human butyrylcholinesterase from frozen Cohn fraction IV-4 by ion exchange and Hupresin affinity chromatography, Plos One, doi.org/10.1371/journal pone.0209795]. 

Comment 2 Fig 4: Judged only from the shown gel it cannot be concluded that the protein is a tetramer, since there is no marker used. There are several high molecular weight marker available. Addition of a marker would be very helpful. 

Response 2 The legend to Figure 4 was modified by including the following information. 

 The pattern of BChE active bands from human EDTA plasma (5 µL) in lanes 1 and 10 provides a well-established reference for the oligomeric state of BChE, where C1 is monomeric BChE at 85 kDa, C2 is a disulfide crosslinked dimer of BChE and albumin [21], C3 is the BChE dimer at 170 kDa, and C4 is tetrameric BChE at 340 kDa. The nondenaturing gel has an intense C4 band for the BChE tetramer and weak bands for BChE isozymes C1, C2, and C3. 

Comment 3 The staining of the gel is derived from the activity of BChE. Tetramers , trimers, dimer and monomeric forms were active. However, in lane 2 (sample 62) there is a distinct band clearly visible below the main band, but this band does not correlate to the trimer, dimer or monomer form. This should at least be mentioned and discussed. 

Response 3 The extra band in lane 2 is consistent with the BChE trimer. This was mentioned at the end of the last paragraph in the section Tetramer status of BChE purified by chromatography on Hupresin.

Comment 4 Is it possible to stain the gel after the reaction with Coomassie blue?

Response 4 It is possible to stain the gel with Coomassie Blue after the gel has been stained for activity. An example of this is given in figure 9 from Schopfer, Lockridge, David & Hinrichs; Purification of human butyrylcholinesterase from frozen Cohn fraction IV-4 by ion exchange and Hupresin affinity; Plos One doi.org/10.1371/journal.pone.0209795 2019.

Comment 5 Lane 572 PBS should be used for phosphate buffered saline. 

Response 5 PBS was substituted for phosphate buffered saline.

Comment 6 It is claimed that the BChE is stable in PBS at 4°C for 20 years. Please give more details on the origin of the samples or Refs. Have the samples all been produced under the same conditions using same reagents?

Response 6 This statement is based on our accumulated experience with BChE produced by various methods. The first sentence of the section “Stability of BChE” has been changed “BChE produced by various methods when stored in PBS, sterile and concentrated (e.g. 6000-7000 u/mL), at 4°C, is stable for up to 20 years [Lockridge Pharmacol Therap 148, 34-46 (2015)].”

Comment 7 Samples shown in Fig 5 are all derived from the new Cohn fraction, correct ?

Response 7 Correct. We have added a statement to that effect in the legend to figure 5.

Comment 8 Stability assessment was done by activity measurements. Has the integrity of the proteins been checked ? e.g. by MS or analytical SEC?

Response 8 No, we have not checked stability in those ways. We have added the following explanation to the second paragraph in the section “Stability of BChE” to expand on this point. Stability was assessed by activity assays. We did not investigate the biochemical changes that occur in the lyophilized BChE protein during years of storage at 27˚C. Stability studies of stored antibodies suggest performance loss can be due to aggregation, fragmentation and oxidation [23].

Comment 9 The BChE prepared from the new Cohn fraction is less stable than the old batches (up to 20 years at 4°C). This should be mentioned and discussed ? Is there a possible reason for this ?

Response 9 There appears to be some confusion here. Purified BChE is stable for 20 years when stored at 4°C, sterile. BChE in the crude new process Cohn Fraction IV-4 starting material is stable for at least 7 years when stored at -80°C. We have modified figure legend 5 to clarify this confusion. “BChE, purified from new Cohn fraction IV-4, was lyophilized in ….”

Comment 10 Kallikrein activity: There are certain differences between different BChE batches. What is the upper limit for kallikrein kinase activity allowed for human preparations – please add value – as done for LPS level (despite this material is not intended for use in humans).

Response 10 The value for kallikrein activity in normal human plasma is 220±30 mU/ml. This value is reported in the BioVision Inc instructions to users of their kit. We have added the value to the text of our paper.

Comment 11 Line 671 “… does not yield pure protein” please give a number.

Response 11 The last paragraph in section “Advantages and disadvantages of affinity chromatography on Hupresin” has been modified to include the statement “It is important to point out that purification of BChE from plasma in a single step with Hupresin yields BChE that is only 10-15% pure [Onder, David, Tacal, Schopfer & Lockridge, Hupresin retains binding capacity for butyrylcholinesterase and acetylcholinesterase after sanitation with sodium hydroxide, Front Pharmacol doi: 10.3389/fphar.2017.00713].”

Comment 12 684. The last sentence is a little bit misleading. The sentence on line 684 states “The new method significantly reduces production costs by optimizing yield and quality of 99% pure butyrylcholinesterase”.

Response 12 This sentence summarizes an important point of our paper. We do not understand how it is misleading.

Comment 13 Is the BChE produced with the new method 99% pure ? Please describe the method for estimating this rate.

Response 13 The following section was added to the Results and Discussion:

 BChE purity

 BChE purified from new process Cohn fraction IV-4 by a single chromatography step on Hupresin affinity resin is 99% pure. Purity was assessed by specific activity using butyrylthiocholine as substrate, where 500 u/mg protein was considered to be 99% pure. Purity was also assessed by contaminating bands seen in SDS PAGE stained with Coomassie blue (Fig 3), where no contaminating bands could be seen.

Comment 14 More analytical data for the purified BChE should be added. E.g. analytical SEC, analytical HIC, MS-Data. HPLC…..

Response 14 MS-data are described in the responses to comments 1 & 13. SEC, HIC and HPLC were not performed.

Comment 15 Maybe it is possible to identify the proline-rich peptide.

Response 15 The following description of the polyproline tetramerization domain was added to the BChE purification” section. “The polyproline peptide (PSPPLPPPPPPPPPPPPPPPPPPPPLP) from RAPH1 (Ras-associated and pleckstrin homology domains-containing protein 1, lamellipodin, gi:82581557) that serves to lock-in the BChE tetramer [Li, Schopfer, Masson & Lockridge, Lamellipodin proline rich peptides associated with native plasma butyrylcholinesterase tetramers; Biochem J 411, 425 (2008)] was identified in the mass spectral data.”

Comment 16 As the protein has been washed with Triton x-100: How can you be sure, that Triton X-100 is completely removed and does not stick to the protein ? I bet it does.

Response 16 Triton X-100 was only used for the solvent/detergent tests. Many of the purifications were performed without Triton X-100. But we agree that it is very hard to completely remove all the Triton X-10 after a protein has been exposed. To address this point, we have added the following to the section “Solvent/Detergent passes through Hupresin without binding”. “It may be difficult to completely remove all of the Triton X-10 after a protein has been exposed. However, in those instances where Triton X 100 was used, there is no evidence for practical consequences from residual Triton X-100 on the final purified BChE. For example, Triton X 100 inhibits BChE [Li, Stribley, Ticu, Xie, Schopfer, Hammond, Brimijoin, Hinrichs & Lockridge, Abundant tissue butyrylcholinesterase and its possible function in the acetylcholinesterase knockout mouse, J Neurochem 75, 1320 (2000)] but the specific activity of the final BChE after purification in the presence or absence of Triton X-100 is the same (see Table 2).”

Minor topics:

Please give sources for “complex” compounds.

Comment 17 Line 176: Source for EDTA

Response 17 “(Sigma-Aldrich cat# E2,629-0, St Louis, MO)” was added

Comment 18 Line 181: Please give conditions for mixing time (rpm, …)

Response 18 “at 60 rpm” was added

Comment 19 Line 276 The abbreviation TMA is used. It should be introduced when the term is used the first time (line 76 tetramethylammonium bromide). Please give source.

Response 19 “(TMA, Tokyo Chemical Industry, cat# T0135, Tokyo Japan)” was added to line 76.

Comment 20 Line 211: source for solvents and detergent (tri-(n-butyl) phosphate, Triton X-100)

Response 20 “Tri-(n-butyl) phosphate, 98% was from Alfa Aesar, cat# A16084, 98%, Wardhill, MA and Triton X100 was from Acros Organics/ThermoFisher cat# 9001 93-1, Pittsburg PA.” was added to line 211.

Comment 21 Line 278: PBS Please explain: is this “self-made” PBS (please give composition) or from a supplier (please give source)

Response 21 “PBS (phosphate buffered saline) was prepared in-house. A 10X solution contained 71.2 g Na2HPO4*2H2O, 9.6 g KH2PO4, 320 g NaCl, and 8 g KCl. After 10-fold dilution this gave 140 mM Na/KCl in 12 mM Na/KPO4 pH 7.4.” was added to the last paragraph in the “Buffer exchange” section.

Comment 22 Lines 318/319: Please give source for dithiobisnitrobenzoic acid, and butyrylthiocholine iodide.

Response 22 “(Sigma Cat# D8130, St Louis MO)” was inserted after dithiobisnitrobenzoic acid.

 “(Fluka/Honeywell cat# 20820, Charlotte NC)” was inserted after butyrylthiocholine iodide.

Comment 23 Line 325 Source for “bovine albumin” which kind ? fatty acid free ?

Response 23 “(Sigma, bovine fraction V, fatty acid free, cat# A8022, St Louis, MO)” was inserted after bovine albumin.

Comment 24 346 SDS polyacrylamide: Which running buffer was used ? Which marker was used ? please give sources.

Response 24 “Running buffer was 25 mM Tris, 180 mM glycine, and 0.01% sodium dodecyl sulfate. Molecular weight markers were Odyssey one-color protein molecular weight markers from LiCor (Lincoln NE) cat# 928-40000.” was inserted at the end of the paragraph in the “SDS polyacrylamide gel” section.

Comment 25 352: Nondenaturing polyacrylamide gel: Which running buffer was used ?

Response 25 “in 22 mM Tris, 23 mM glycine pH 9” was inserted at the end of the first sentence in the “Nondenaturing polyacrylamide gel stained for BChE activity” section.

Comment 26 428: Citation should be in the reference section. Use number for ref. only

Response 26 citation Holloway, Anal Biochem 53, 304 (1973) has been moved to the reference section.

---

## [Editor Report · Decision Letter 1]

28 Dec 2022

Human butyrylcholinesterase in Cohn fraction IV-4 purified in a single chromatography step on Hupresin

PONE-D-22-24553R1

Dear Dr. Schopfer,

We’re pleased to inform you that your manuscript has been judged scientifically suitable for publication and will be formally accepted for publication once it meets all outstanding technical requirements.

Kind regards,

Ajaya Bhattarai

Academic Editor

PLOS ONE

Additional Editor Comments (optional):

The revised manuscript looks good.

Reviewers' comments:

<quillbot-extension-portal></quillbot-extension-portal>

---

## [Editor Report · Acceptance letter]

4 Jan 2023

PONE-D-22-24553R1 

Human butyrylcholinesterase in Cohn fraction IV-4 purified in a single chromatography step on Hupresin 

Dear Dr. M. Schopfer:

I'm pleased to inform you that your manuscript has been deemed suitable for publication in PLOS ONE. Congratulations! Your manuscript is now with our production department. 

Kind regards, 

on behalf of

Dr. Ajaya Bhattarai 

Academic Editor

PLOS ONE